# Do Stop Me Now: Detecting Boilerplate Responses with a Single Iteration

## Abstract

Large Language Models (LLMs) often expend significant computational resources generating boilerplate responses, such as refusals, simple acknowledgements and casual greetings, which adds unnecessary cost and latency. To address this inefficiency, we propose a simple yet highly effective method for detecting such responses after only a single generation step. We demonstrate that the log-probability distribution of the first generated token serves as a powerful signal for classifying the nature of the entire subsequent response. Our experiments, conducted across a diverse range of small, large, and reasoning-specialized models, show that the first-token log-probability vectors form distinctly separable clusters for different response types. Using a lightweight k-NN classifier, we achieve high accuracy in predicting whether a response will be a substantive answer or a form of boilerplate response, including user-specified refusals. The primary implication is a practical, computationally trivial technique, optimizing LLM inference by enabling early termination or redirection to a smaller model, thereby yielding significant savings in computational cost. This work presents a direct path toward more efficient and sustainable LLM deployment.

## 1 Introduction

Large Language Models (LLMs) have revolutionized artificial intelligence with their ability to understand and generate human-like text across diverse applications, from conversational agents to code assistants. This remarkable capability comes with a significant challenge: the high computational and financial costs associated with llm inference for each user query (Regmi & Pun, 2024). These costs are especially wasteful in real-world scenarios where LLMs generates unwanted or predictable outputs. For instance, OpenAI CEO Sam Altman has publicly stated that politeness expressions like "please" and "thank you" have cost the company tens of millions of dollars, due to the electricity consumption associated with generating these boilerplate responses (USA Today, 2025). This highlights a critical inefficiency: LLMs often produce unnecessary tokens that consume resources without contributing to the user's core intent. The ability to accurately and cost-effectively characterize LLM responses *prior to or early in their generation* is thus paramount for optimizing inference costs, reducing latency, and enhancing the overall sustainability of LLM-powered systems.

To address this inefficiency, recent research purposes several novel methods, specifically concerning refusal-to-comply (i.e. due to safeguards). For example, strategies like refusal tokens (Jain et al., 2025) involve prepending special tokens during training that the model learns to generate first when a refusal is appropriate. Other studies show that LLMs encode global attributes of their future responses in hidden representations even before any tokens are generated, enabling emergent response planning (Dong et al., 2025). Another study (Arditi et al., 2024), reveals that refusal behavior is classifiable by a one-dimensional subspace in LLM activations, leading to the development of a "refusal metric". This metric, derived from summing probabilities assigned to specific "refusal tokens" at early generation stages, serves as an efficient proxy for measuring refusal likelihood without full response generation, aligning closely with our objectives for cost-effective content filtering.

This work builds upon these advancements and shows that the **log-probabilities of the first generated token** are sufficient for accurate **prediction of multiple response types**, including user-specified refusals. By focusing on early prediction and detection of boilerplate content (such as refusals, gratitude acknowledgements and other non-task-solving elements), we aim to significantly

reduce the computational and environmental footprint of LLM inference, ultimately leading to more efficient and sustainable AI systems.

## 2 PRELIMINARIES

### 2.1 DEFINING BOILERPLATE RESPONSES

We define boilerplate responses as responses that are interchangeable within their class. For example, **refusal type response** (i.e. *"I'm sorry, I cannot help you with ___"* or *"You're welcome! I'm glad I could assist you with ___"*) are interchangeable up to context. Efficient and cost-effective identification of boilerplate enables optimizing inference costs in real-world applications. This involves distinguishing content such as refusals, gratitude, or simple acknowledgements from genuinely meaningful, task-solving outputs. Recent advancements in LLM research offer several avenues to address this challenge, often by leveraging the internal mechanisms of these models for early prediction or classification.

## 3 RELATED WORK

### 3.1 DIALOGUE ACT CLASSIFICATION FOR CONVERSATIONAL INTENT

In conversational settings, Dialogue Act Classification (DAC) identifies the communicative intent behind utterances (Zhangwenbo & Yuhan, 2025; Aljanaideh, 2025). LLMs exhibit zero-shot DAC capabilities, which can be refined iteratively with online feedback without requiring labeled data (Zhangwenbo & Yuhan, 2025). This allows for the early classification of dialogue acts, including those that signify conversational fillers, gratitude, or simple acknowledgements, thereby identifying non-meaningful conversational turns.

### 3.2 DISTINGUISHING MEANINGFUL CONTENT FROM BOILERPLATE ELEMENTS

A key challenge is the precise differentiation between essential, reasoning tokens and repetitive, non-critical boilerplate tokens (e.g., formatting, transitional phrases like *"Based on the user's request..."*) (Ye et al., 2025). The Shuffle-Aware Discriminator (SHAD) offers an automated and adaptive solution by exploiting predictability differences after shuffling input-output combinations: boilerplate tokens remain predictable, while reasoning tokens do not.

### 3.3 OPERATIONAL EFFICIENCY METHODS FOR COST REDUCTION

Beyond direct classification, other techniques aim to reduce LLM inference costs by avoiding unnecessary computation. Semantic caching mechanisms, such as GPT Semantic Cache, leverage query embeddings to identify semantically similar questions and retrieve pre-generated responses, significantly reducing API calls and improving response times, especially for repeated boilerplate queries (Regmi & Pun, 2024). Additionally, advancements in multi-token prediction enable LLMs to jointly predict several subsequent tokens in a single inference step (Orgad et al., 2025). While this primarily speeds up generation, it could potentially be adapted to quickly scan for and flag boilerplate patterns, allowing for early termination. Another approach is to use model routing, sending simpler prompts (such as those leading to boilerplate responses) to a smaller, cheaper model (Ding et al., 2024).

### 3.4 EARLY PREDICTION OF OUTPUT CHARACTERISTICS

A foundational concept in this area is emergent response planning, where LLMs' internal hidden representations can encode global attributes of an entire future response before any tokens are generated (Dong et al., 2025). By probing these pre-generation representations, it is possible to predict various characteristics of the upcoming output, such as response length, reasoning steps, answer confidence, or factual consistency. This capability is instrumental for pre-generation resource allocation optimization, allowing systems to anticipate the nature of a response and potentially avoid

costly full generations of boilerplate content. Complementary to this, methods like TRAIL (Shahout et al., 2025) leverage recycled LLM layer embeddings to dynamically predict remaining output length with low overhead and high accuracy, refining predictions at each token generation step. This approach, similar to others employing separate lightweight LLMs or BERT models (Devlin et al., 2019) for length prediction, aims to optimize scheduling and reduce latency, indirectly supporting early termination for short, non-meaningful responses.

## 3.5 REFUSAL DETECTION AT INFERENCE TIME

For specific boilerplate types like refusals, specialized mechanisms have been developed. The Refusal Tokens strategy proposes prepending a special `[refuse]` token (or category-specific tokens) to responses during training, allowing the model to learn to generate this token first when a refusal is appropriate (Jain et al., 2025). At test-time, the softmax probability of this refusal token quantifies the likelihood of a refusal, enabling calibrated control over refusal rates without retraining. This enables a "cheap sweep" by allowing identification of optimal refusal thresholds with a single forward pass, avoiding full response generation. Notably, this method allows for fine-grained control over various refusal types if multiple tokens are used.

Similarly, (Arditi et al., 2024) demonstrates that refusal behavior in LLMs is mediated by a one-dimensional subspace within their residual stream activations. This allows for the derivation of a "refusal metric" by summing the probabilities assigned to a predefined set of "refusal tokens" (e.g., "I'm sorry", "I cannot") at the first token position of the prompt. This metric serves as an efficient proxy for estimating the likelihood of a model refusing an instruction without requiring full response generation. This approach is the most closely related to our work, as it focuses on leveraging early signals from the model's generation output to predict the nature of the upcoming response. Yet, unlike our work, it requires manual listing of these "refusal tokens".

Beyond explicit refusals, research into LLM transparency also explores their internal signals for other content characteristics. Studies on LLM hallucinations indicate that truthfulness information is concentrated in "exact answer tokens" within the generated response (Orgad et al., 2025). Probing classifiers trained on intermediate representations of these tokens can predict errors, suggesting that LLMs encode information about their own truthfulness. While focused on error detection, this highlights the general potential to probe internal states for the "meaningfulness" of a response.

## 4 OUR METHOD

When using LLMs to generate responses, they do so one token at a time. At each iteration, *all possible tokens* are assigned probabilities, but only one is selected. Thus, examining all token probabilities of a *single iteration* provides an overview of all the possible subsequent responses. We hypothesize that by using the (log-)probabilities of the first token generation, it is possible to classify certain response types. To validate it, we measure the similarities between log-probabilities of the first token generated by different prompts, designed to induce boilerplate responses versus others that suppose to elicit more detailed responses related to the chat. We perform these validation over different models, both small and large language models, as the token-probability-space is not comparable between different models.

### 4.1 DATASET

We create a unique dataset of ~3k different chats of different lengths and different classes. We define several types of classes:

- **Refusal**: Chats or messages an assistant will refuse to answer due to internal learnt safeguards.

- **Thanks**: Chats ending with the user thanking the assistant for its assistance. These usually make the assistant respond with common phrases like *"You're welcome!"* or *"My pleasure!"*

- **Hello**: Chats starting with the user saying *"Hello!"*, *"Hi!"* or similar texts.

- **Chat**: All other chats that do not belong to above classes, where the user and the assistant are having a regular conversation.

The dataset was created in the following way:

1. We use the AdvBench dataset (Zou et al., 2023), containing harmful prompts, and classify these as Refusal.

2. We then sampled ~500 random prompts from the Alpaca dataset (Taori et al., 2023), and classify these as Chat. The input-prompts of the Alpaca dataset are split into two columns: *instruction* and *input*, where in most cases, the *input* column is empty. We explicitly used only the *instruction* column as the prompt, thus creating some cases of chats with missing context.

3. Per each of the prompts we now have, we prompted an LLM to respond, and recorded the response. We then asked an LLM to continue the conversation as the user would, and if the model refused to reply - steer the conversation to a legitimate follow up question. We combined the User-Assistant-User interactions as new examples, and labeled them as Chat.

4. For the original harmful prompts, we asked an LLM to hypothesize what was the legitimate prompt the user asked *prior* to the harmful prompt. We then asked an LLM to respond the the legitimate prompt, and added these User(safe)-Assistant-User(harmful) interactions as Refusal.

5. Next, we asked an LLM to come with 250 "benign" thank-you prompts a user might send to an assistant (i.e. *"Thank you for your help!"*). We then sampled 500 User-Assistant-User chats, replaced the last message with a random thank-you prompt, and labeled them as Thanks.

6. Finally, we created a list of ~30 "benign" prompts which can be used to initiate a conversation with an assistant without any additional request (such as *"Hello!"* or *"Good morning!"*). We labeled these as Hello.

The result is a dataset of ~3k chats, containing both single prompt chats and User-Assistant-User chats. The dataset is available at `ANON_URL`[1].

## 4.2 RESPONSE TYPE CLUSTERING AND CLASSIFICATION

We prompt selected language models with the chats from our dataset, and then record the log-probabilities vectors of the first token generated as the LLM's response. We visualize the results using 2D t-SNE (van der Maaten & Hinton, 2008). To quantitatively evaluate the effectiveness of our approach, we trained k-Nearest Neighbors (k-NN) classifiers (Cover & Hart, 1967) on the first-token log-probability embeddings for each model category. The k-NN algorithm was chosen for its simplicity and interpretability, enabling direct measurement of the clusters separability. For each model, we performed 5-fold stratified cross-validation to ensure robust evaluation across all response types (Chat, Hello, Refusal, Thanks). We fixed $k = 3$ for all models to enable direct comparison across different architectures. All reported metrics (accuracy, precision, recall, F1-score) are cross-validation results averaged across the 5 folds, providing more reliable estimates than single train-test splits. We report macro-averaged precision, recall, and F1-score to account for class imbalance, particularly for the Hello class which represents only ~1.4% of the dataset.

## 5 EXPERIMENTS

We verify our hypothesis on three scenarios: **Small Language Models**, **Reasoning Models** and **Large Language Models**.

### 5.1 SMALL LANGUAGE MODELS

We perform an evaluation of the first-token log-probabilities of the following models:

---

[1]Dataset is attached to anonymous submission, dataset link is hidden for anonymity

- Llama 3.2 3B (Meta AI, 2024)
- Qwen 2.5 1.5B (Qwen Team, 2024; Yang et al., 2024)
- Gemma 3 1B (Gemma Team et al., 2025)

A 2D t-SNE plot of these models' log-probabilities is shown in Figure 1 and Table 1. We clearly see a separation between the classes.

### 5.1.1 REFUSAL DUE TO INCAPABILITY

When examining Chat samples located much closer to the Refusal class, we find mostly the chats with the missing context we created by omitting the *input* column from the Alpaca dataset. The following list provides a few examples of such incomplete chat messages found more closely to the Refusal class:

- *"Based on the provided input paragraph, provide a summary of its content."*
- *"Translate the sentence below into Japanese."*
- *"Describe the character of the protagonist in the given TV show."*
- *"Group the given list into 3 Groups."*

The assistants are now incapable of replying to these prompts, as they now miss critical context. They therefore trigger a refusal response from the assistants, as they try to explain to the user that they cannot assist - not due to safeguards, but due to lack of context.

### T-SNE plots of first-token log-probs of SLMs

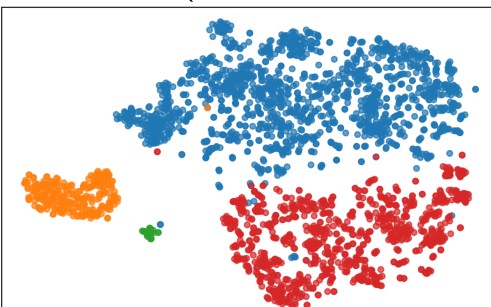
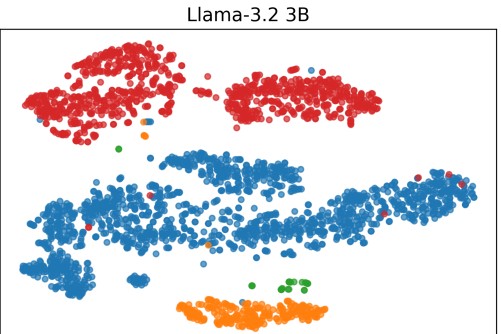

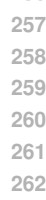

Figure 1: 2D T-SNE plot of first-token log-probabilities for Small Language Models (Llama 3.2 3B, Qwen 2.5 1.5B, Gemma 3 1B). Each point represents a chat, colored by class.

Table 1: Small Language Models Performance on Type Classification (k=3)

| Model | Accuracy | Precision | Recall | F1 | Chat F1 | Hello F1 | Refusal F1 | Thanks F1 |
|---|---|---|---|---|---|---|---|---|
| Qwen2.5-1.5B | 0.997 | 0.991 | 0.998 | 0.994 | 0.998 | 1.000 | 0.998 | 0.998 |
| Llama-3.2-3B | 0.995 | 0.996 | 0.984 | 0.990 | 0.998 | 1.000 | 0.998 | 0.996 |
| Gemma-3-1B-IT | 0.994 | 0.997 | 0.997 | 0.997 | 0.998 | 1.000 | 0.997 | 1.000 |

### 5.1.2   REFUSAL DUE TO SYSTEM PROMPT

So far, chats marked as Refusal are those AI assistants tend to refuse to reply to due to internal trained safeguards. We wish to check whether this method applies too when the refusal is due to an arbitrary system prompt, stating that the assistant cannot reply to certain scenarios.

In the experiment shown in Figure 2, we send the assistant a request for a recipe for a Black Forest Cake. We did so twice per model, once with a system prompt explicitly stating that the assistant *cannot* provide a recipe for a Black Forest Cake, and once without any additional restrictions. Figure 2 shows the same 2D T-SNE plot of the first-token log-probabilities of SLM, along with the log-probabilities of the first token of each of the Black Forest Cake prompt variations. We clearly see that when the model was instructed not to provide a recipe, the first token log-probabilities are closer to the Refusal class center-of-mass. While not shown here, other such experiments yielded similar results.

T-SNE plots of first-token log-probs of SLMs, including the Cake Recipe experiment

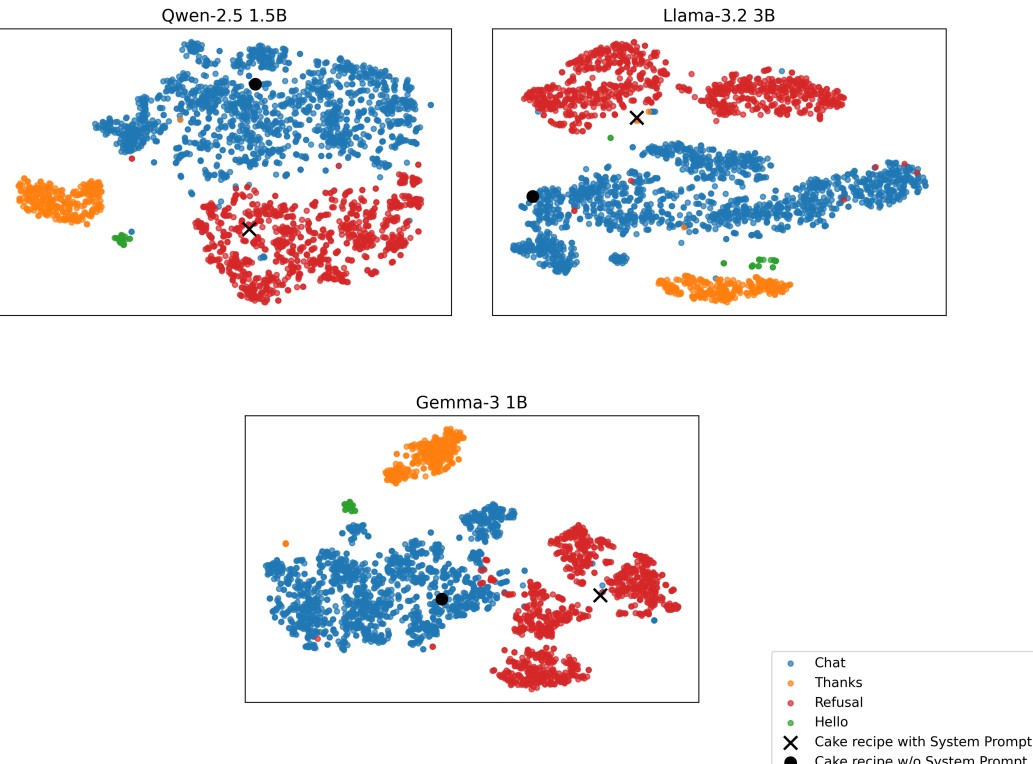

Figure 2: 2D T-SNE plot of first-token log-probabilities for Small Language Models, including the Cake Recipe experiment. The black circle represents a request for a recipe for a Black Forest Cake, and the black cross represents the same requests, but with a system prompt instructing the assistant not to provide the recipe.

## 5.2 Reasoning Models

The examination of reasoning models adds a bit more complexity, as they tend to begin with a thinking phase which includes self-explaining and "self-talking" (i.e. *"So, the user asked...", "I need to...",* etc.). Since our classification applies to the first token of the **response**, we slightly adjusted the chat inputs by adding an empty-thinking phase to the assistant's message (i.e. *"\<think>\</think>"*), and only then checked the log-probabilities of the first token.

We evaluated the following reasoning models:

- DeepSeek-R1 8B (Llama Distillation) (DeepSeek-AI, 2025)
- Phi-4 Reasoning Plus (Abdin et al., 2025)

Results are shown in Figure 3 and Table 2. Here too, we see a clear separation between the classes.

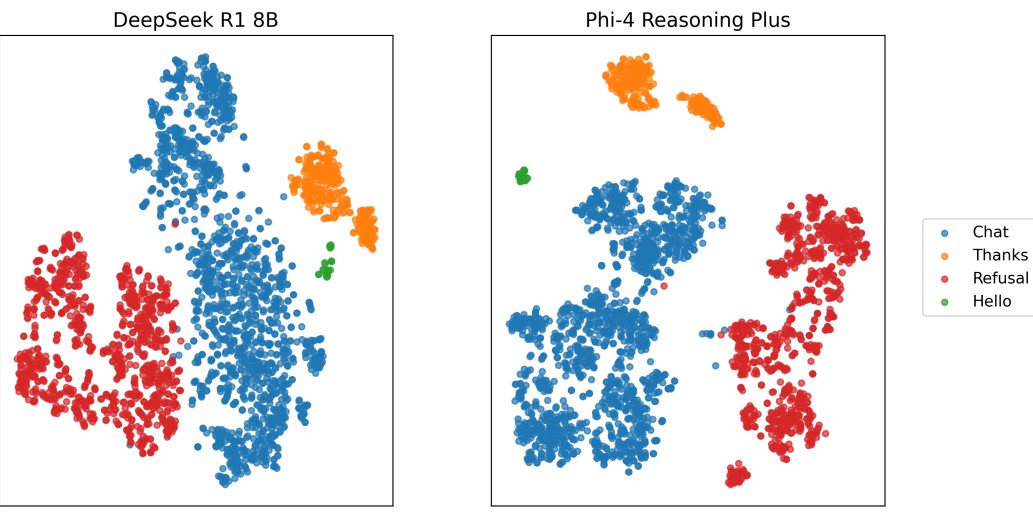

Figure 3: 2D T-SNE plot of post-empty thinking (*"\<think>\</think>"*) first-token partial log-probabilities for Reasoning Models (DeepSeek-R1 8B, Phi-4 Reasoning Plus). Each point represents a chat, colored by class.

Table 2: Reasoning Models Performance on Type Classification (k=3)

| Model | Accuracy | Precision | Recall | F1 | Chat F1 | Hello F1 | Refusal F1 | Thanks F1 |
|---|---|---|---|---|---|---|---|---|
| Phi-4-Reasoning+ | 0.998 | 0.999 | 0.999 | 0.999 | 0.999 | 1.000 | 0.999 | 1.000 |
| DeepSeek-R1-8B | 0.998 | 0.998 | 0.989 | 0.993 | 0.999 | 1.000 | 0.999 | 1.000 |

## 5.3 Large Language Models

We perform an evaluation of the first-token log-probabilities of the following cloud-based models:

- OpenAI GPT-4o[2] (OpenAI et al., 2024)
- Gemini 2.0 Flash (Hassabis & Kavukcuoglu, 2024)

Unlike open-source models, these models do not provide the full log-probabilities of generated tokens, but only the top 20. We reconstructed the partial log-probabilities vectors, as the APIs of

---

[2]Running on Microsoft Azure

these models provide both the log-probability and token IDs (token indices) of the top 20 tokens.[3] Therefore, the vectors displayed in Figure 4 and Table 3 are *trimmed* log-probabilities vectors.

Even with the trimmed log-probabilities, we still see clear separation between the classes.

### T-SNE plots of first-token partial log-probs of LLMs

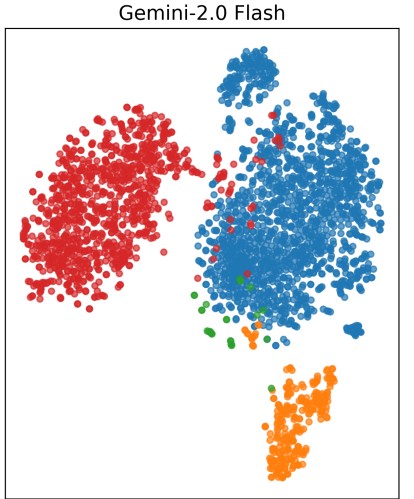
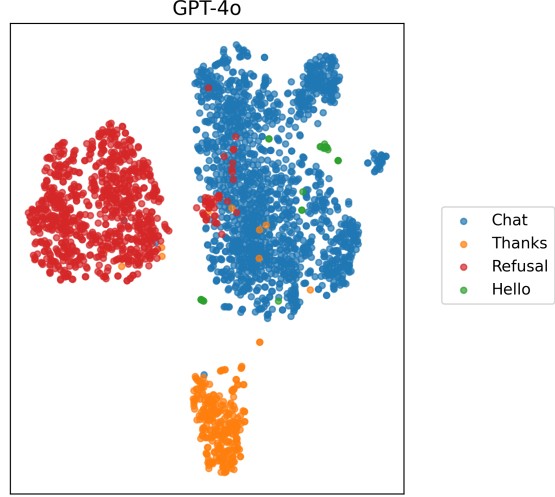

Figure 4: 2D T-SNE plot of first-token partial log-probabilities for Large Language Models (GPT-4o, Gemini-2.0 Flash). Each point represents a chat, colored by class.

Table 3: Large Language Models Performance on Type Classification (k=3)

| Model | Accuracy | Precision | Recall | F1 | Chat F1 | Hello F1 | Refusal F1 | Thanks F1 |
|---|---|---|---|---|---|---|---|---|
| Gemini-2.0-Flash | 0.979 | 0.989 | 0.844 | 0.884 | 0.993 | 0.826 | 0.995 | 0.993 |
| GPT-4o | 0.974 | 0.983 | 0.914 | 0.941 | 0.992 | 0.941 | 0.987 | 0.982 |

## 6 CONCLUSION

In this work, we addressed the significant computational inefficiency of Large Language Models generating boilerplate responses. We introduced a simple yet highly effective method to predict the nature of an entire response by analyzing the log-probability distribution of just the first generated token.

Our comprehensive experiments across a diverse range of small, large, and reasoning-specialized models consistently demonstrated that the log-probabilities of the first token form distinct, separable clusters for different response types, such as substantive answers, refusals, simple acknowledgements or greetings. We showed that a lightweight k-NN classifier can leverage these clusters to achieve high accuracy in predicting the response category after a single generation step. Furthermore, our method successfully identifying refusals prompted by both inherent model safeguards and arbitrary, user-defined system prompts.

The primary implication of our findings is a practical and computationally trivial technique to optimize LLM inference. By enabling early termination of unwanted boilerplate generation, this approach offers substantial savings in computational cost and latency. This work presents a direct path toward more efficient, economical, and sustainable deployment of LLM systems, paving the way for more responsive and cost-effective applications.

---

[3]For GPT-4o, we used `tiktoken` to retrieve the token IDs from the token string-values: *github.com/openai/tiktoken*

Future work could focus on applying this technique to a wider range of boilerplate categories, multi-language scenarios, and exploring its effectiveness in multi-modal contexts.

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
