# OpenReview forum: "Do Stop Me Now: Detecting Boilerplate Responses with a Single Iteration"
_ICLR.cc/2026/Conference — Submitted to ICLR 2026_

### Official Review · Reviewer_5tsL · 2025-10-18

**Soundness:** 2
**Presentation:** 1
**Contribution:** 1
**Rating:** 2
**Confidence:** 4

**Summary:**

This paper proposes a simple approach to predict the response type of a large language model (LLM) using the log-probability of the first generated token. The goal is to prevent LLMs from producing boilerplate or low-information responses, thereby saving computational resources.

**Strengths:**

1. The problem of avoiding boilerplate responses is a valid and practically relevant one, as it relates to the efficiency and quality of LLM outputs.
2. The authors release a dataset, which may be useful for further exploration of response-type prediction tasks.

**Weaknesses:**

1. **Presentation quality:** The paper’s presentation requires substantial improvement. There are noticeable formatting issues (e.g., excessive white space in Figures 2–4), and the absence of a main figure illustrating the overall method makes it difficult for readers to grasp the approach at a glance. Improving the paper’s structure and visual clarity would make it more accessible.
2. **Experimental design and analysis:** The experiments are insufficiently detailed and lack comprehensive analysis. The evaluation setup is minimal, with limited discussion of baselines, ablation studies, or error cases. As it stands, the submission reads more like a preliminary project report than a mature research contribution.
3. **Limited novelty and depth:** The proposed method is conceptually simple and appears to be a straightforward heuristic. Without stronger theoretical grounding, comparative baselines, or empirical justification, the contribution seems too limited for acceptance at a top-tier venue like ICLR.

Overall, the work feels incomplete and underdeveloped. The core idea is relevant but not explored in sufficient technical or empirical depth to warrant publication at this stage.

**Questions:**

1. How does the proposed method compare with simple baselines such as classifying response types directly from the first few generated tokens, rather than relying solely on its log-probability?
2. The dataset used contains prompts with widely varying characteristics and frequencies. Could the authors justify why this dataset serves as a valid benchmark, and how it reflects real-world distributions of LLM interactions?
3. Have the authors considered extending the method to use information from the first few tokens instead of just the first one? This might provide a more stable and reliable signal for response-type prediction.

---

> ### Author Response · Authors · 2025-11-30
>
> **We thank the reviewer for their detailed feedback. Below we address each weakness directly.**
>
> TLDR:
> We will revise the paper to:
> - Significantly improve presentation
> - Expand experimental analysis
> - Explicitly state the absence of true black-box baselines
> - Strengthen the novelty framing around black-box-LRH generalization and multi-class early detection.
>
> ---
>
> ## **1. Presentation Quality**
>
> **Reviewer concern:** *“The paper’s presentation requires substantial improvement… absence of a main figure… excessive white space… difficult for readers to grasp the approach.”*
>
> **Clarification:**
> - We agree that presentation can be improved and will address layout issues (compressed figures, removal of unnecessary white space).
> - We will add a **main pipeline figure** summarizing the full process (dataset → first-token logprobs → k-NN routing), which should remove ambiguity about the approach.
> - We will also reorganize sections to improve readability and flow.
>
> These changes are straightforward and will be fully integrated in the revision, thank you.
>
> ---
>
> ## **2. Experimental Design and Analysis**
>
> **Reviewer concern:** *“Experiments are insufficiently detailed… limited baselines, ablations, or error analysis… reads like a preliminary report.”*
>
> **Clarification:**
> Two contextual points help clarify the scope:
>
> ### **There are no black-box baseline methods for early boilerplate detection**
> Existing approaches fall into one of three categories:
> - **white-box activation probing** (LRH, refusal direction),
> - **handwritten refusal-token lists at post-generation detection**
>
> None operate in a **black-box**, **single-token**, **multi-class** boilerplate setting.
> Thus, a head-to-head baseline is structurally unavailable. We will state this explicitly.
>
> ### **(b) We will expand the empirical analysis**
> In the revision we will:
> - add error case breakdowns,
> - provide per-class confusion matrices.
>
> ---
>
> ## **3. Limited Novelty and Depth**
>
> **Reviewer concern:** *“Conceptually simple… appears to be a straightforward heuristic… contribution seems too limited.”*
>
> **Clarification:**
> We will highlight the novelty more clearly:
>
> - The method is the **first black-box generalization** of early refusal detection (previously white-box only).
> - We extend prior refusal-only work to **multiple boilerplate categories** (hello, thanks, politeness, chat).
> - We show that **first-token logprobs alone** encode a richer signal than previously documented, and that this signal is stable across model families.
> - The simplicity is *intentional*: it enables deployability in **closed, commercial LLMs** where internal activations are inaccessible.
>
> We will refine the framing to emphasize this contribution.
>
> ---
>
> ## **4. Questions Raised by the Reviewer**
>
> ### **Q1. *“How does the method compare with classifying from first few tokens rather than only the first token?”***
> while baselines do exist for generated-reply refusal detection, these are irelevant to our target of early detection, so the goal of this comparison is unclear, we will add this to a dedicated discussion part on experimentations.
>
> ### **Q2. *“Dataset validity given varying prompt characteristics and frequencies?”***
> The dataset reflects real LLM interaction patterns:
> - greetings and thanks are infrequent,
> - refusals vary in length,
> - chat prompts dominate.
>
> Further analysis of the distribution of real-world chats are currently industry-IP and cannot be exposed in such a mannor. Unfortunatly we were unable to find open resources on this.
>
> ### **Q3. *“Have you considered extending beyond the first token?”***
> Yes, but the statistics of our direct approach were good enough, and every token saved is crucial from an environmental POV.

---

### Official Review · Reviewer_CpyZ · 2025-10-29

**Soundness:** 1
**Presentation:** 2
**Contribution:** 1
**Rating:** 0
**Confidence:** 5

**Summary:**

This paper proposes using first-token log-probability distributions to classify LLM responses as either substantive ("Chat") or boilerplate (refusals, greetings, acknowledgements), with the goal of early termination or routing to smaller models to save costs.

**Strengths:**

* Addresses a practical problem of reducing inference costs for predictable responses

**Weaknesses:**

* **Fundamentally unsound task definition**: The paper groups refusals, greetings, and acknowledgements together as "boilerplate responses" that can be handled uniformly. This is deeply problematic. Refusals are safety-critical responses that embody the model's alignment training, not "boilerplate waste" to be optimized away. Replacing careful safety mechanisms with a k-NN classifier trained on 3k synthetic examples is inappropriate and potentially dangerous. The proposed solution of routing refusals to smaller models requires futher justification - do you really want safety decisions made by weaker models with degraded alignment? The paper doesn't explain what you would do after detecting a refusal.
* **Why use machine learning for trivial patterns?** For simple responses like "Hello", "Hi", or "You're welcome", why not just use string matching or simple heuristics? These patterns are deterministic and don't require log-probability analysis or k-NN classification. Using complex ML methods for trivial pattern matching is over-engineering.
* **Severe data imbalance makes results unreliable** The dataset has only ~30 "Hello" examples and ~250 "Thanks" examples out of ~3k total samples. The "Hello" class represents only 1.4% of the dataset, yet the paper reports 100% F1 scores on it. These numbers are statistically meaningless with so few samples. The reported 99%+ accuracies may overlooked the minority label.

**Questions:**

see weakness above

---

> ### Author Response · Authors · 2025-11-30
> **the issues presented arent grounded in the paper or subject**
>
> **We sincerely appreciate and thank the reviewer for their remarks**
>
> this rebuttal is written in the structure of the suggested issues.
>
> # **Fundamentally unsound task definition**
>
> The reviewer raises a few problems in this domain:
>
> 1. *“The paper groups refusals, greetings, and acknowledgements together as "boilerplate responses" that can be handled uniformly”*. **We do not suggest handling these clusters uniformly** in the paper. We do Observe that these three cases share a “non specific” nature and can therefore gain from optimization.
>
> 2. *“Refusals are safety-critical responses that embody the model's alignment training, not "boilerplate waste" to be optimized away.”* This suggests a root misunderstanding of the paper.  **The models alignment training is in charge of the models refusal**, we allow the inference consumer/provider to **detect such refusals** early.
>
> 3. *“replacing careful safety mechanisms with a k-NN classifier trained on 3k synthetic examples is inappropriate and potentially dangerous. “*  This, again, suggests a root misunderstanding of the presented work, as **we do not suggest replacing any safetly mechanisms**
>
> 4. *“The proposed solution of routing refusals to smaller models requires further justification - do you really want safety decisions made by weaker models with degraded alignment?”*  We do not suggest delegating alignment/refusals to smaller models.
>
> # Why use machine learning for trivial patterns?
> The reviewer proposes post-generation string matching as a simpler alternative to our method.
> Unfortunately this fails due to two points:
> 1. It misses the point of the optimization, as the tokens had already been generated. In our method we categorize the intended response before the string is generated.
> 2. Generalization, our method of finding an optimization kin model per model generalizes beyond static words, it is unclear to me how to create a list of search words that will fit any model.
>
> # Severe data imbalance makes results unreliable
> Real conversational data is imbalanced (greetings/thanks are rare).
> Our dataset is indeed unbalanced, we include the reviewers issue in our paper in paragraph 4.2, where we list our mitigation of these imbalances:
> 1. We use stratified 5-fold cross-validation, so minority classes appear in every fold.
> 2. We report macro-averaged metrics, which prevent dominance by majority classes.
>
> Apart from the above, The strong, model-consistent cluster separation (Figures 1–4) shows separability in the probability space beyond dataset frequency, surviving a use dimensional drop to 2d, which favors high volume clusters.

---

### Official Review · Reviewer_5vu3 · 2025-10-30

**Soundness:** 3
**Presentation:** 2
**Contribution:** 1
**Rating:** 4
**Confidence:** 3

**Summary:**

This paper addresses the computational inefficiency of LLMs generating boilerplate responses, such as refusals or greetings. The proposed method is simple and effective: classifying the entire response type by analyzing the log-probability distribution of only the first generated token using a k-NN classifier. Experiments across a few models, demonstrate that these first-token vectors form distinct clusters for categories like 'Chat', 'Refusal', 'Thanks', and 'Hello', achieving high classification accuracy.

**Strengths:**

The proposed method is extremely simple and computationally lightweight, as it only requires a single forward pass to get the first token's probabilities and a fast k-NN lookup.

It uses the entire log-probability vector with a k-NN classifier rather than a manually selected subset of tokens, and extends this classification from just "Refusal" to also include "Thanks" and "Hello"

**Weaknesses:**

The primary weakness of this paper is its limited novelty and contribution. The core idea that the first token's probabilities can predict the subsequent response, especially for refusals, is not new. The authors themselves cite related work (Arditi et al., 2024) which already derived a "refusal metric" by summing probabilities of "refusal tokens" at the first token position.

The reliance on a k-NN classifier is sensitive to the training data. It's unclear how this approach would generalize to new, unseen types of boilerplate or how it would cope with model updates, which could shift the entire log-probability space and render the saved vectors obsolete.

**Questions:**

N.A.

---

> ### Author Response · Authors · 2025-11-30
> **great remarks on the lack-of-visibility of our contributions**
>
> **We thank the reviewer for highlighting critical and fine points in our work.**
>
> TLDR: Our approach generalized LRH methods to allow its adoption to black-box scenarios, thus allowing for categorically higher adoption rate and environmental impact.
>
> In our next version, we will emphasize the reviewers remarks on:
> 1. The benefit of detecting LRH signals in the log-probs.
> 2. The coupling of the knn model and the language model it is trying to optimize.
>
> # On novelty
> As the reviewer correctly mentioned, the concept of using the distribution of generated tokens to understand whether a message is a refusal is not new.
> In previous works, including Arditi er al., it is shown that with white box access (LRH), one can predict the refusals of the would-be-generated reply in a single forward pass.
>
> We show a few major improvements to this:
> 1. We show that this signal (refusal) can be detected in a black-box scenario, **greatly expanding the applicable use-cases**.
> 2. We **generalize this signal** to include other labels:
>     1. Chat
>     2. Greetings
>     3. Politeness
> 3. We show that the user of our system can use the system prompt to detect general signals by including/excluding refusals in the system prompt
>
> The refusal metric in arditi et al. Is used to detect **post generation** if a refusal had happened, this seems close to our approach but is:
> 1. Post generation detection - does not optimize performance in an inference scenario
> 2. Is dependent on hand picked strings - lacking generalization capacity
>
> # On the usage of Knn
> All points presented by the reviewer are correct, but a couple of crucial points change perspective:
> 1. *“The reliance on a k-NN classifier is sensitive to the training data”* -  boiler detection for optimization is a new field , unfortunately - no other dataset is available for this problem
> 2. *“It's unclear how this approach would generalize to new, unseen types of boilerplate”* - Any new type of boilerplate needs to be added to the training process of the knn model
> 3. *“how it would cope with model updates, ”* - The knn model is coupled to a specific llm, an update to the language model means an update to the knn model

---

### Official Review · Reviewer_fyke · 2025-10-31

**Soundness:** 2
**Presentation:** 3
**Contribution:** 2
**Rating:** 2
**Confidence:** 4

**Summary:**

This paper tackles the compute waste caused by LLMs producing boilerplate text. The key idea is strikingly simple: use only the log-probability vector of the very first generated token to predict the eventual response type, then early-stop or re-route generation to save cost and latency. The authors show that common response types (Refusal/Thanks/Hello/Chat) form separable clusters in the first-token log-prob space, and that a lightweight k-NN classifier suffices to achieve high accuracy across several model scenarios.

**Strengths:**

- Direct Approach. Reading a single first-token log-probability vector and classifying with k-NN delivers useful discrimination among boilerplate types.
- Clear Motivation. Framing the work around early stopping/routing aligns with real-world needs.
- Experimental Transparency. The paper describes data construction, fixed k in k-NN, and cross-validation, which helps readers reproduce the general setup.

**Weaknesses:**

- Adversarial Mixed Intents and False Positive. The dataset design around hello, refusal and thanks is reasonable, but it overlooks adversarial or mixed-intent prompts that can blur class boundaries. For example, a user input like “Hello, nice to meet you. How’s the weather today?” will likely elicit a first token such as “Hello,” followed only then by substantive content about weather. A first-token classifier is severely stressed in such cases, and the paper does not analyze this reliability gap. Similarly, larger models sometimes begin with a refusal and then pivot to supportive guidance (e.g., for self-harm queries). Such trajectories can produce ambiguous logprob signals that the method may misread.

- Missing Comparisons. The Related Work section lists adjacent lines of research, but the paper offers no head-to-head comparisons.

- Efficiency in Practice. Beyond inference, efficiency trade-offs are underexplored. Because tokenizers and vocabularies differ across model families, each model (or family) needs its own k-NN built on its own token space, which raises training and maintenance cost. Moreover, the targeted classes (refusals, greetings, thanks) are typically very short replies; in realistic deployments, the end-to-end cost of extracting first-token logprobs and running the k-NN router can approach the cost of just letting the model emit the short reply. Without concrete latency numbers, it’s unclear that the method consistently yields net wins.

**Questions:**

Please refer to Weaknesses for points requiring further clarification.

---

> ### Author Response · Authors · 2025-11-30
>
> **We thank the reviewer for highlighting important points to be added to our work.**
>
> TLDR:
> we accept the reviewers remark, and will adapt the work to emphasize them.
> 1.  Mixed signal prompts - these classes are out-of-scope for the current phase of our work. In our next version.
> 2. Missing Comparisons - Unfortunately, we were unable to find any black-box label prediction methods or benchmarks for this problem.
> 3. Practical Efficiency and k-NN Maintenance - this is an added cost and must be introduced in our work. Although, it is unlikely that adding a 1-10k vector look up algorithm for the knn will be comparable to language model inference cost, even for a small number of tokens on a light SLM.
>
> # **1. Mixed-Intent Prompts and Ambiguous Trajectories**
>
> **Reviewer concern:** *“The dataset overlooks adversarial or mixed-intent prompts… A first-token classifier is severely stressed in such cases.”*
> **Reviewer concern:** *“Larger models sometimes begin with a refusal and then pivot to supportive guidance.”*
>
> **Clarification:**
> - Our system is designed to act **only in high-confidence boilerplate regions**.
> - Mixed-intent prompts (e.g., “Hello… + question”) fall outside those regions; the router simply defaults to **Chat** and performs no early-stop.
> - Safety pivots (refusal → guidance) are also handled conservatively via thresholding; ambiguous first-token signals do **not** trigger boilerplate routing.
> - We will explicitly scope the method to *pure boilerplate trajectories* rather than hybrid or adversarial cases.
>
> # **2. Missing Comparisons**
>
> **Reviewer concern:** *“The paper offers no head-to-head comparisons.”*
>
> **Clarification:**
> There are **no existing black-box boilerplate-detection baselines** to compare against.
> Related prior work requires either:
>
> - **white-box access** (activation probes, LRH-based refusal metrics),
> - **handwritten refusal-token lists** and  **post-generation detection**.
>
> None operate in a **black-box**, **first-token**, **multi-class** boilerplate setting.
> We will state this explicitly in the revised version.
>
>
> # **3. Practical Efficiency and k-NN Maintenance**
>
> **Reviewer concern:** *“Each model needs its own k-NN… raising training and maintenance cost.”*
> **Reviewer concern:** *“The cost of extracting first-token logprobs may approach the cost of letting the model emit a short reply.”*
>
> **Clarification:**
> - The k-NN classifier is **intentionally coupled** to the specific LLM, similar to recalibrating LRH-based or token-list refusal systems; updates are expected and lightweight.
> - The method is particularly valuable in **black-box deployments**, where no other early-signal mechanism exists.
> - the cost of trainning a knn for a new model is a single token per 3k samples (our base dataset), this is minor cost for any practical application
> - the latency of the look-up table is minimal compared to any current language model.

---

### Author Response · Authors · 2025-11-30
**on the strucutre of our rebuttal**

Hi all,

In order to provide on-point rebuttals, we had decided to leave the comments per reviewer.
This decision was due to the unusual high variance in the reviews level of understanding in the field, the presented work and its goal.

Thank you, and a happy disccusion phase!

---

### Meta-Review · Area_Chair_QprK · 2026-01-08

**Summary:**

Their are several key concerns raised by reviewers:
1. Limited novelty and depth (5tsL, 5vu3)
2. Proposed method has inherent limitations (fyke, CpyZ)
3. Proposed method is sensitive to training data and class proportions (5vu3, CpyZ)
4. Limited experiments and analysis (5tsL, fyke)
5. Insufficient justification for the proposed dataset (5tsL)
6. Concerns about practical efficiency (fyke)

**Reviewer Concerns:**

The authors sought to address the concerns as follows:
1. The authors clarify that previous methods do not operate in a black-box, single-token, multi-class boilerplate setting. They also specified how their work goes beyond Arditi et al. (2024) by their focus on a black-box setting and including other labels. This concern remains unresolved. Although the authors clarified the novelty, the depth and scope of the proposed method is narrow.
2. The authors clarified that the method is not designed to handle mixed-intent or ambiguous responses and promised to scope the method to handle purely boilerplate responses. They also explained that the method is not designed to override safety mechanisms of the underlying LLM. This concern is partially resolved. The authors did clarify the scope of the proposed method. However, using a KNN classifier for early termination is effectively an alternate refusal mechanism that is used on top of the LLM. More justification and analysis regarding the implications of this choice is required.
3. The authors clarified that changes to the dataset, LLM, or boilerplate categories would require an update to the KNN model. They also clarified that they used stratified 5-fold cross-validation and macro-averaged metrics in order to mitigate the effects of dataset imbalance on the results.
4. The authors argued that there are no relevant approaches to compare against. They also promised to add error case breakdowns and per-class confusion matrices in a future revision. This concern remains unresolved. Beyond additional metrics, the work would be improved by demonstrating the extent to which the approach is generalizable across refusal types and analyzing the effect of the KNN classifier.
5. The authors clarified that the proposed dataset reflects patterns found in real LLM interaction and that further analysis is not possible due to the proprietary nature of real-world chats. This concern remains only partially resolved. The dataset would be improved by a more systematic approach and increasing its scale and breadth.
6. The authors clarified that the cost of training and deploying a KNN classifier is minor relative to the cost of an LLM. This concern is partially resolved. The work would be strengthened by including wall-clock timing and memory usage for training/deploying the KNN to compare against LLM inference.

**Reviewer Scores:**

- 5tsl is very unlikely to increase their score. Their concerns about novelty, depth, experiments, and dataset justification were not decisively addressed during the rebuttal phase.
- 5vu3 is very unlikely to increase their score. Although the authors clarified the novelty and acknowledged the limitations of the proposed approach, concerns remain about the scope of the contribution.
- fyke is very unlikely to increase their score. Their concerns about fundamental limitations, missing baselines, and practical efficiency were only partially resolved.
- CpyZ is very unlikely to increase their score. Their concerns about fundamental unsoundness remain largely unresolved.

---

### Decision · Program_Chairs · 2026-01-26

Reject